# Global and Local Processing of Letters and Faces: The Role of Visual Focal Attention

**DOI:** 10.3390/brainsci13030491

**Published:** 2023-03-14

**Authors:** Silvia Primativo, Lisa S. Arduino

**Affiliations:** Department of Human Science, LUMSA University, Piazza delle Vaschette 101, 00193 Rome, Italy

**Keywords:** local/global processing, letters processing, faces processing, focus attention

## Abstract

Studies in the literature have shown how the preference towards local or global processing can vary according to different characteristics of the stimuli involved, such as stimulus type and stimulus time duration. In the present study, we investigated whether letters and faces undergo similar or different global/local processing and the attentional mechanisms that might be linked to eventual differences. We used hierarchical, congruent, and incongruent letters and faces in different time conditions (180 and 500 ms) and we conducted three different experiments. The results of Experiment 1 showed that with stimuli shown for 180 ms, letters are processed more efficiently at the local level, with an inversion of the global interference effect. Conversely, faces are still processed more efficiently at the global level, with evidence of global advantage and global interference. The results of Experiment 2 showed that when the same stimuli are presented for longer (500 ms), they are still processed differently. Indeed, we observed faster local processing for letters but still a tendency, even if not significant, toward a global processing advantage for faces. Moreover, the cue-size effect, i.e., the ability to modulate visual focal attention based on the characteristics of the cue, was measured. In Experiment 3, the cue-size effect showed a statistically significant correlation with the local processing advantage for letters but not for faces. We conclude that during the almost automatic processing of letters it is possible to modulate focal attention on the basis of the task, narrowing the field of visual attention during the local task and neglecting the global stimulus. Conversely, during face processing, the holistic mechanism tends to prevail over focal attention modulation skills, even when it is explicitly required to focus on the local stimulus.

## 1. Introduction

Object perception is a complex task for the human visual system. One aspect that has received much attention in the literature is whether visual processing proceeds from the global aspects of an object to the local ones or from the local to the global.

In 1977, Navon [1] highlighted a system tendency towards global processing in the first instance during a complex object perception task. In his seminal study, the author used hierarchical compound stimuli consisting of large letters (the global level) composed of small letters (the local level). Global and local letters could be similar or different (e.g., a large E composed of local E’s or local S’s). The participant’s task was to respond to one of the levels while ignoring the other unattended level or vice versa. A global advantage, denoted as the global precedence effect (GPE), has been demonstrated in such a task, under a wide variety of conditions [1,2,3]. The GPE has been described as characterized by two main components: (i) a global advantage effect (GA), consisting of a faster reaction time (RT) and a higher accuracy at identifying the global level of the stimulus; and (ii) a larger global interference effect (GI) in the processing of incongruent stimuli from the global to the local element. Since these two effects (GA and GI) have often been reported together in the literature [4,5,6,7], it was assumed that they were a valid measure of the order in which local and global levels of structure are processed, supporting the view that global level precedes local level processing.

However, further evidence questioned this assumption, in that it was shown that the preference for global or local processing could be a function of the nature of the stimuli [7]. In particular, variations in stimulus characteristics, such as stimulus size and stimulus duration, may result in shorter RTs for the local compared to the global level and a reduction or a reversal of the global interference [4,5,7,8,9,10,11]. In terms of stimulus size, Kinchla and Wolfe [9] found a global advantage in reaction time with patterns subtending less than approximately 7° of visual angle and a local advantage with larger patterns. According to Navon and Norman [6], this inverted effect might be caused by globality being confounded with eccentricity (i.e., distance from the fovea). Indeed, with larger hierarchical letters, the global letter is farther from the fovea than some of the local letters, meaning that the local letter can benefit from greater acuity.

Not only the stimulus size and typology but also the stimulus duration have been listed among the factors impacting the interference effect, even though some results are contradictory [2]. Indeed, Paquet and Merikle [12] presented compound letters for 10, 40, or 100 ms and found global-to-local interference only at the shortest exposure duration. At longer exposure durations, mutual interference effects were observed. However, in other studies, the global advantage has been observed at longer exposure durations as well [2,11,12,13,14].

Taken together, these results challenge the hypothesis that the global advantage and the global interference reflect the order of a coarse-to-fine temporal dynamic processing in visual object recognition, suggesting that the initial processing strategy may depend on several factors. Miller [15], for example, proposed a model in which the detection of global and local information takes place in parallel and becomes available for decision processing with the same time course. Though, subsequently, he suggested that decision and response selection processes operate in a global-to-local fashion.

The hypothesis that local and global levels of information are processed in parallel is also supported by neuropsychological evidence, suggesting that there are two separate mechanisms associated with different cortical regions [16,17]. However, which factors and mechanisms are determinant in pushing for a local or a global advantage is still a question of debate. A further variable which may influence the GA and GI effects is the type of stimulus, and this represents the main focus of the present study.

At first, we wanted to further corroborate the finding that GA and GI are related to different factors and mechanisms. It has been shown that the GA disappeared when global stimuli, consisting of Navon’s letters, subtended from 7 to 10° of the visual angle [9]; but does this hold true also for faces? In this study, we used both of these types of visual objects with a prediction that with bigger stimuli we would observe a local level advantage in RT and that such local preference would affect letters but not faces. Indeed, Fink et al. [18] showed the activation of different brain circuits during the global and local processing of letter-based and object-based hierarchical stimuli. The authors concluded that hemispheric activations during global and local processing are modified by the stimulus category. With the aim of investigating the global interference effect, Poirel et al. [19] adopted meaningful and meaningless hierarchical stimuli and showed that the global interference effect was evident only for meaningful stimuli, such as letters and objects, but not for meaningless non-objects.

In this study, we go further by making the assumption that letters and faces are differently sensitive to the local and global level of analysis, and that, as reported in the literature, when the letters are bigger, an RT advantage for the local level might be expected, whereas this might not be true for faces. The second hypothesis we tested relates to the fact that some studies have shown that the GPE can be reduced or even reversed not only by the stimulus physical characteristics but also by exposure duration in the direction that a longer duration (100 vs. 40 ms) reduces the GPE [20,21], in terms of both the RT advantage and the interference effect.

Here, we wanted to investigate whether the reduction of the interference effect due to longer stimulus duration is valid not only for letters, as shown in the literature, but holds true for faces as well. We made the prediction that the global interference would be more robust with faces, even in longer durations. The third goal of the present study was to investigate whether an attentional focus mechanism might play a role in explaining some of the differences between letter and face processing. Indeed, focal attention is a component of visuospatial attention and refers to the ability to adjust the size of the attentional window according to the dimensions of the stimulus to be processed [22,23,24,25,26,27,28,29]. Such an adjustment favors higher efficiency in the processing of the material within the attended area [30]. Usai et al. [31] reported that a shift of attention is observed when attending to local features, while an expansion of the focus of attention takes place when attending to global features. We wanted to test the hypothesis according to which attentional focus is more “adjustable” for some types of stimuli than for others, eliciting a larger (holistic) or smaller (local) processing, which will influence the participant’s performance with specific task requirements. In this hypothesis, the nature of the stimulus would move the participant’s attentional focus toward a more restricted or a more enlarged attentional window.

In order to test our hypotheses, in this study, perceptually matched hierarchical letters and faces were used as stimuli. Letters are simple and overlearned perceptual elements, and they are easily recognized globally by expert readers, but, for discriminative purposes, they might require a local shift of attention, for example to differentiate between the letter R and the letter P. We hypothesized that, in the case of letters, participants should be able to adjust their attentional focus to the global or local element according to the required task (identification of the global or the local element). In particular, in the local identification task, a narrower focusing of spatial attention [32] should elude strong interference effects. Conversely, face processing is primarily holistic in nature [33]. In this case, it might be hypothesized that it would be more difficult to adjust and restrict attentional focus to the local element, thus causing a larger global interference during a local identification task.

To summarize, this study was led by three main hypotheses, that: (1) the local advantage would be larger for letters than faces; (2) the global interference effect would be larger in face vs. letter processing, even at longer stimulus duration; and (3) the ability to adjust and modulate the attentional focus would correlate with letter but not face processing. In order to test these hypotheses, we conducted three experiments. In Experiment 1, participants had to process face and letter hierarchical stimuli presented for 180 ms. In Experiment 2, the same stimuli were presented for 500 ms. Finally, in Experiment 3, we explored the participant’s ability to modulate their visual focal attention as a function of the stimulus’ features, in order to test whether letters and faces may benefit differently from this manipulation. The prediction was that the higher efficiency in modulating the size of the attentional window would positively correlate with a local advantage in letter but not in face processing.

Experiment 1—Processing of hierarchical stimuli presented for 180 ms.

## 2. Method

### 2.1. Participants

Twenty-three healthy participants were recruited and tested at the Department of Human Science, LUMSA University, Rome. Participants (M:F = 6:17) had an average age of 24.5 years (SD = 4.4) and an average education level of 14.7 years (SD = 2.2 years).

### 2.2. Apparatus

Participants were seated in front of a 15-inch computer monitor with a resolution of 1920 × 1080 pixels and a refresh rate of 60 Hz. The viewing distance was constant at 57 cm. The experiment was controlled by the SR Research Experiment Builder software (SR Research Ltd., Kanata, ON, Canada). The same apparatus was used for all the experiments described below.

#### 2.2.1. Experiment 1a—Hierarchical Letter Processing

In Experiment 1a, we used hierarchical stimuli (see Figure 1) that consisted of global patterns of three different letters (E, H, and S, in the Courier New font style), formed from smaller local patterns of the same letters. All possible combinations of congruent and incongruent stimuli (i.e., identity of the global and local shapes was the same or different, respectively) were applied, resulting in a total number of nine different combinations. The overall stimulus size was 15 × 15°, the local letter’s size was 1.2 × 1.2°, and the interletter distance was 0.6° (border to border).

Two separate experiments were run for global and local tasks. For the local task, the participant was asked to report the local or small letter, and, for the global task, the participant was required to report the global or big letter. The order of presentation of the two conditions (similarly with the two conditions of Experiment 2) was counterbalanced across participants. Each condition (global or local) consisted of 90 trials: 30 congruent, and 60 incongruent. Participants were first familiarized with a printed version of the stimuli, and the experimental procedure was started only when the participant correctly responded to three consecutive items in both the global and the local tasks.

After the familiarization phase, the experiment was initiated. A fixation cross subtending 0.5° of visual angle appeared in the middle of the screen for 300 ms, then disappeared and the stimulus appeared for 180 ms. The participant’s task was to report either the global or the local letter, and to respond as fast and accurately as possible. The participant was asked to respond by pressing a key on a keyboard (patches were placed on the three H, J, and K keycaps of the keyboard, indicating the letters H, S, and E, respectively). Accuracy and response time were measured.

#### 2.2.2. Experiment 1b—Hierarchical Faces Processing

In Experiment 1b, we used faces as hierarchical stimuli (see Figure 2) to investigate global and local processing. The stylized global face could be happy, sad, or surprised. Similarly, the local faces that composed the global face could be happy, sad, or surprised. As for Experiment 1a, all possible combinations of congruent and incongruent stimuli were used. The stimulus size was the same as for the letters: 15 × 15° for the global face and 1.2 × 1.2° for the local letter. Additionally, the same interletter distance was used (0.6°). The stimuli were created by using the Photoshop software. Participants were first familiarized with the printed version of the stimuli, and the experimental procedure was started only when the participant correctly responded to three consecutive items in both the global and the local tasks.

After the familiarization phase, the experiment was initiated. A fixation cross subtending 0.5° of visual angle appeared in the middle of the screen for 300 ms, then disappeared and the stimulus appeared for 180 ms. The participant’s task was to report either the global or the local face emotion and to respond as fast and accurately as possible. The participant was asked to respond by pressing a key on a keyboard (patches were placed on the three H, J, and K keycaps of the keyboard, indicating that the stimulus was happy, sad, and surprised, respectively. Accuracy and response time were measured.

## 3. Results

A repeated measures ANOVA was run on reaction times with congruency (congruent vs. incongruent), task (global vs. local), and stimuli (faces vs. letters) as within-subject factors. The SPSS 26 software (IBM Corp, 2019; IMB SPSS Statistics for Windows, Version 26.0 Armonk NY) was adopted to run the statistics. *p*-values equal or below 0.05 were considered as statistically significant and were further investigated by direct *t*-tests comparisons. Effect sizes were evaluated and are reported as partial eta squared values (ηp2).

The results indicated a main effect of congruency [F(1, 22) = 12.1; *p* = 0.002; ηp2 = 0.342), with shorter RTs for congruent (625 ms) vs. incongruent stimuli (643 ms). Crucially, the triple interaction of congruency x task x stimuli was statistically significant [F(1, 22) = 6.8; *p* = 0.02; ηp2 = 0.226] and is represented in Figure 3. The interaction revealed that faces but not letters required longer processing times in the local task when there was incongruency between the global and the local stimulus, as compared to when these were congruent (*p* < 0.001). This did not hold true for letters (*p* = 0.67). While the overall RTs for processing faces and letters were similar (647 vs. 621; *p* = 0.063), the difference between the two stimuli only emerged in the local processing of incongruent stimuli, with faster processing times for letters compared to faces (593 vs. 682 ms; *p* < 0.0001).

### 3.1. Interim Conclusions

The results seem to suggest that participants can segregate more efficiently the local element of letters, which is not affected by the global interference. Conversely, the global processing of faces seems to be inevitable, impacting the processing of local elements and slowing it down in the case of incongruency between the local and global elements.

Experiment 2—Processing of hierarchical stimuli presented for 500 ms.

### 3.2. Participants

Twenty-four healthy participants were recruited and tested at the Department of Human Science, LUMSA University, Rome. Participants (M:F = 10:14) had an average age of 23.1 years (SD = 3.5) and an average education level of 14.7 years (SD = 1.7 years). Demographic features of the two groups of participants were matched between Experiments 1 and 2 (*t*-test comparisons; all *p* > 0.1).

#### 3.2.1. Experiment 2a—Hierarchical Letter Processing

The same design, stimuli, and procedure of Experiment 1a was adopted. The only difference was that stimulus presentation was set to 500 ms.

#### 3.2.2. Experiment 2b—Hierarchical Face Processing

The same design, stimuli, and procedure of Experiment 1a was adopted. The only difference was that stimulus presentation was set to 500 ms.

### 3.3. Results

The main results are reported in Table 1. When the stimuli were presented for 500 ms, we observed a significant main effect of the stimulus type (F(1, 23) = 10.8, *p* = 0.003; ηp2 = 0.32), better clarified by the significant interaction of stimulus x congruency (F(1, 23) = 10.02, *p* = 0.004; ηp2 = 0.303) and stimulus x task (F(1, 23) = 16.3, *p* < 0.0001; ηp2 = 0.414). The two interactions are reported in Figure 4.

The first interaction revealed that, at 500 ms, both faces and letters undergo a congruency effect, with longer response times in case of incongruency between the global and the local elements.

Furthermore, the stimulus x task interaction revealed that the local task is faster for letters (604 vs. 644; *p* = 0.01) but not for faces (683 vs. 663; *p* = 0.2; see lower panel of Figure 4). Crucially, and differently from Experiment 1 (stimulus presentation of 180 ms), the triple interaction of congruency x task x stimuli did not reach statistical significance (F(1, 23) = 0.7, *p* = 0.4; ηp2 = 0.029).

## 4. Interim Discussion

The results of Experiment 2 evidenced, as in Experiment 1, faster RTs for local vs. global for letters but not for faces, further suggesting that participants can more efficiently segregate the local element of letters but not of faces. Moreover, the interaction of stimulus x congruency suggests that, in the case of long available times for the processing of the stimuli, the presence of an incongruency slows down reaction times in both directions: the incongruent global element not only impacts the processing of the local element, but the incongruent local element also slows down the processing of the global element.

Experiment 3—Focal attention test.

In the focus test, we explored the participant’s ability to modulate visual focal attention as a function of the stimulus’ features. In order to do so, we measured the effect of different cues on target detection. Following the procedure used in Albonico et al. [22], we measured RTs with three conditions and a baseline: (1) a red dot as an optimal cue for the target position but not the focal component, since it does not convey any information about the size of the target stimulus; (2) a small square as an optimal cue for the focal component, since it encloses the target stimulus without masking it and conveys spatial information about the optimal field of integration necessary to detect the target stimulus; and (3) a big square as a non-optimal cue for the focal component, because it directs the focus to a spatial area much larger than the target stimulus. A baseline condition by which the target appearance was not pre-cued was also included. Crucially, the cues’ physical features matched the features of the stimuli used in Experiments 1 and 2. Indeed, the small cue had the same size of the local element of the hierarchical letters and faces, and the big cue had the same size of the global element of the hierarchical letters and faces. Stimulus onset asynchrony (SOA), that is, the interval between the appearance of the cue and that of the target, was manipulated in order to explore both the exogenous and the endogenous allocation of attention [34]. Indeed, according to Epstein et al. [34], shorter SOAs evoke an exogenous and automatic orienting of attention, while longer SOAs elicit a more voluntary and endogenous orientation of attention.

### 4.1. Participants

All participants who took part in Experiments 1 and 2 were also administered the focus test. Overall, 47 participants (M:F = 16:31), with an average age of 23.8 years (SD = 4.0) and an average education level of 14.7 years (SD = 2 years), were tested.

### 4.2. Stimuli and Procedure

The target stimuli consisted of a capital letter T (font style Sloan, color black) of 1 × 1° of visual angle, oriented upright, while the cue was represented by a red dot (diameter of 0.4°), by a small black square (1.2 × 1.2°, line thickness = 0.1°), by a big black square (15 × 15°, line thickness = 0.1°), or by no cue. Both the target stimulus and the cue were displayed on a gray background. Compared to Albonico et al. [22], only the foveal condition was run so that both the cue and the target were presented at the center of the screen. The participants had to detect the presence of the target stimulus on the screen by pressing the space bar on a keyboard with their right index finger. The appearance of the target was preceded by one of the three possible cues or by the absence of any cue (baseline), and the SOA between the cue and the target was 100, 300, or 500 ms. The order of the cue type and SOA were randomized within the experiment. Each trial started with a blank gray screen, followed, after 1000 ms, by one of the three possible cues. As in the original study by Albonico et al. [22], we chose not to display any fixation point in order to avoid giving an additional or confounding cue to participants. After the disappearance of the cue, the T target was presented and remained on the screen until the participant responded (or for a maximum of 2000 ms), after which the target disappeared and the next trial started. With this condition, both the target stimuli and cues were always presented at the center of the screen. An example of the experimental procedure is reported in Figure 5. Every participant completed 72 experimental trials. Thirteen catch trials were also included, with cues not followed by the target. In this case, the participant was not supposed to press any key. Trials were randomized across participants and equally divided between the 100, 300, and 500 ms SOA conditions, as well as between the four cue conditions.

### 4.3. Results

The results are reported in Table 2. A repeated measures ANOVA was run with SOA and cue as within-subject factors. The results indicated the main effects of the SOA (F(2, 44) = 37.8; *p* < 0.001), cue (F(3, 43) = 27.8; *p* < 0.001), and a statistically significant interaction of SOA x cue (F(6, 40) = 3.9; *p* = 0.004). The interaction indicated that the cue size effect (faster reaction times when the target was cued by the small square compared to the big square) was observed only in the 300 ms SOA condition (14 ms; *p* = 0.01). The effect was in the predicted direction in the 100 and 500 ms SOA conditions (7 and 5 ms, respectively); however, the difference did not reach statistical significance. At this point, we aimed to investigate whether there was a significant correlation between the cue size effect and the faster processing of local vs. global stimuli. To do so, in the first instance, we calculated two speed indexes, separately for letters and faces. These indexes were measured, for each participant, as the average reaction time in the incongruent global task minus the average reaction time in the incongruent local task, separately for faces and for letters. These indexes represented the acceleration in processing of local vs. global stimuli.

We hypothesized a positive correlation between the acceleration in processing local letters and the cue size effect, whereby the local task might elicit a restriction of the focal attention and an acceleration in the processing of local elements. Conversely, we hypothesized a lack of such correlation between the cue size effect and local face processing. Indeed, we expected that the global, holistic processing of faces is stronger and more automatic, and, therefore, less available for local processing.

As hypothesized, the results indicated a statistically significant correlation between the cue-size effect at the 500 ms SOA and a faster processing of local vs. global letters (*r* = 0.3; *p* = 0.041). None of the cue-size effects showed statistically significant correlations with the faster processing of local vs. global faces (see Table 3).

### 4.4. Interim Conclusions

In the third experiment, we explored visual focal attention by adopting visual cues matched with the local and global elements of the hierarchical stimuli in terms of size. In particular, the small cue had the same size of the local element in the hierarchical stimuli (of both faces and letters) and the large cue had the same size of the global element in the hierarchical stimuli (of both faces and letters). The results seem to suggest that the ability to adjust attentional focus is significantly and positively correlated with the local advantage in letter but not face processing. This is in accordance with our hypothesis that in the case of letter processing the local task might elicit a restriction of the focal attention and an acceleration in processing local elements. This is, instead, unlikely, in case of face stimuli, for which global, holistic processing is predominant and difficult to silence.

## 5. General Discussion

In the present study, we investigated whether different stimulus types, such as letters and faces, undergo similar or different global/local processing and the attentional mechanisms that might be linked to eventual differences. Our research design was led by three main hypotheses, that: (1) the local advantage would be larger for letters than faces; (2) the global interference effect would be larger in face vs. letter processing, even at longer stimulus durations; and (3) the ability to adjust and modulate the attentional focus would correlate with letter but not face processing.

The results of Experiment 1 showed that when stimuli are big and are shown for 180 ms, letters are processed more efficiently at the local level, with an inversion of the global interference effect. Conversely, faces are still processed more efficiently at the global level, with evidence of global advantage and global interference.

The results of Experiment 2 showed that the same large stimuli presented for longer (500 ms) are still differently processed. Indeed, we observed faster local processing for letters but still a non-significant tendency toward a global processing advantage for faces. Moreover, the congruency effect persisted for faces and re-appeared for letters at the longer duration. However, the lack of a triple interaction in this case indicates that not only did the global level interfere with the processing of the local level but also that the local level interfered with the processing of the global level. Finally, in Experiment 3, we showed a statistically significant correlation between the cue-size effect at the 500 ms SOA and faster processing of local vs. global letters, while none of the cue-size effects was statistically significant in correlation with faster local vs. global processing of faces.

Overall, in this study, we demonstrated that the stimulus nature and the cognitive processes it elicits are crucial in preferential global vs. local processing. In particular, we argue that (1) the coarse-to-fine temporal dynamic processing hypothesis in visual object recognition cannot be generalized to stimuli of a different nature; (2) the stimulus size variable is strongly modulated by the stimulus nature; and (3) stimulus duration might have different impacts on the processing of different stimulus types.

Indeed, we showed that there is a faster processing of local letters and global faces and a strong interference of the global on the local only for faces. This is in accordance with the hypothesis advanced by Miller [15] that the detection of global and local information takes place in parallel and becomes available to decision processing with the same time course, with the subsequent decision and response selection processes operating, according to this model, in a global-to-local fashion.

This is also in accordance with Heinze and Munte [35], who conducted an event-related brain potential (ERP) study, adopting a divided-attention paradigm by which subjects were asked to respond to hierarchically structured letter stimuli containing a target letter, either at the global or at the local level. The ERP analysis revealed early posterior negative components (denoted as N250). The N250 components to global and local targets exhibited a different time course and a different topographical distribution, suggesting that they are determined by separate processing structures and that global and local target perception may be mediated by separate brain systems acting, at least initially, in parallel.

In this study, we go further and suggest that the initial parallel processing of global and local features could be modulated by a stimulus nature fostering a different adaptation of the window of focal attention, in terms of restriction or enlargement. Focal attention indeed appears to be modulated by exogenous factors, such as the dimensions and features of the stimulus to be processed or the specific task requirements [22,23,24]. We argue that, in the case of hierarchical stimuli, the local task triggers a restriction of the attentional window size, but only for letter stimuli, which are particularly suitable for local processing [36,37]. In the case of faces, the stimulus nature strongly triggers a holistic processing [33], which makes it more difficult to restrict the focus of attention. Indeed, differently from letters, in the case of faces, the global advantage persists for large stimuli. This is consistent with the results of Harel, Kravitz, and Backer who showed, by using fMRI, that visual object processing is strongly influenced by the observer intent in terms of the activation of brain circuits [38].

Some limitations of the study are worth noting. In particular, the participants’ sample might be improved in future studies, both in terms of size (including larger number of individuals) and demographic features. In the present study, we only included young university students. Indeed, further studies are necessary in order to confirm the generalizability of the present results in different populations (e.g., older individuals and, eventually, clinical populations). In particular, the results from the present study might be further explored and eventually shed some light on the cognitive mechanisms damaged in individuals with specific difficulties in global but not local processing (e.g., simultanagnosia) and in individuals with face processing deficit. On this vein, future studies will also be aimed at generalizing the results of the present study by including one or more control condition adopting different typologies of stimuli, such as geometric shapes. This will strengthen and further support the conclusion that the observed effects are specific to letters and faces.

Our data also seem to suggest that the stimulus duration might have different impacts on the processing of different stimulus types. The literature has already shown that a longer stimulus duration might reduce or even reverse the GPE effect [20,21,35]. Our results confirmed our hypothesis, according to which the stimulus duration effect is, itself, modulated by the nature of the stimulus. Indeed, for a longer stimulus duration (180 ms), the GPE effect was inversed for letters but not for faces, suggesting that the global precedence effect is more robust in the latter case. We also tested the extremely long stimulus duration of 500 ms. The results obtained with this timing showed not only that the interference effect persisted for faces but that it also re-emerged for letters. However, and crucially, the interference in this duration was bidirectional: the global interfered with the local and the local interfered with the global. We interpret this result as the consequence of an interaction between global and local processing, particularly in terms of decision and response selection, rather than perceptual and attentional mechanisms.

In conclusion, we showed that visual stimuli processing is modulated by the nature of the stimuli and that this is strictly linked with the attentional mechanisms required by different stimuli. Letters elicit a local processing, which is mediated by a smaller focus of attention, while faces elicit a more holistic, global processing, with a smaller impact on specific task requirements.

## Figures and Tables

**Figure 1 brainsci-13-00491-f001:**
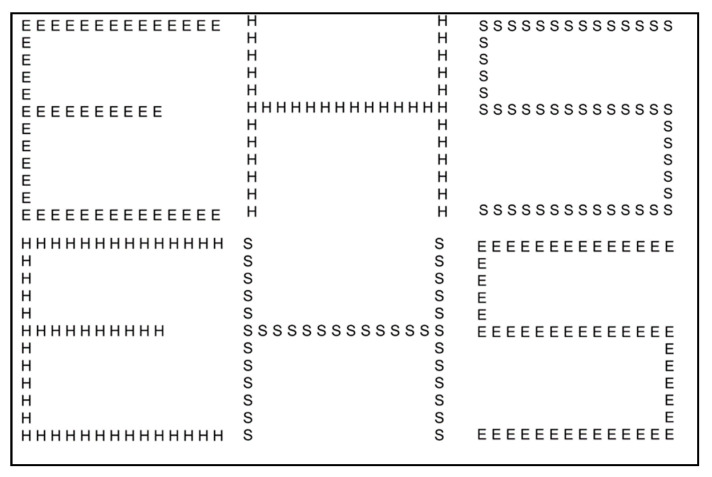
Hierarchical letters. Congruent stimuli are reported in the upper line, incongruent stimuli are reported in the lower line.

**Figure 2 brainsci-13-00491-f002:**
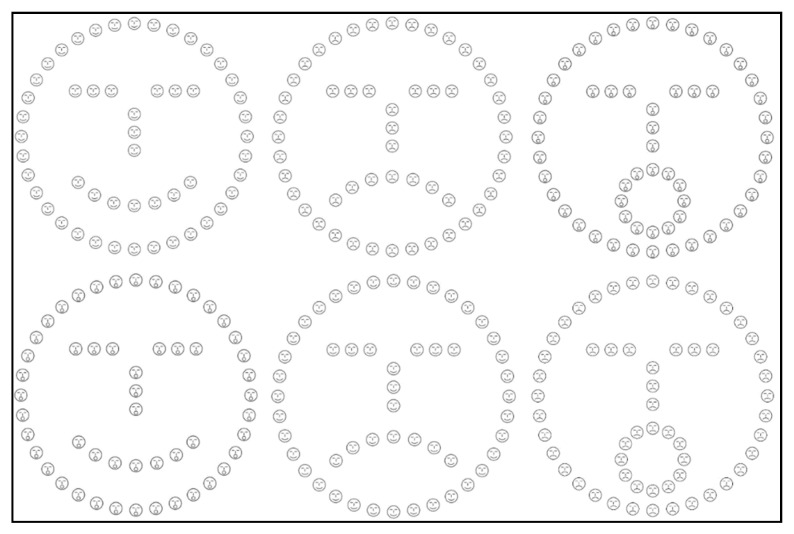
Hierarchical faces. Congruent stimuli are reported in the upper row, incongruent stimuli are reported in the lower row.

**Figure 3 brainsci-13-00491-f003:**
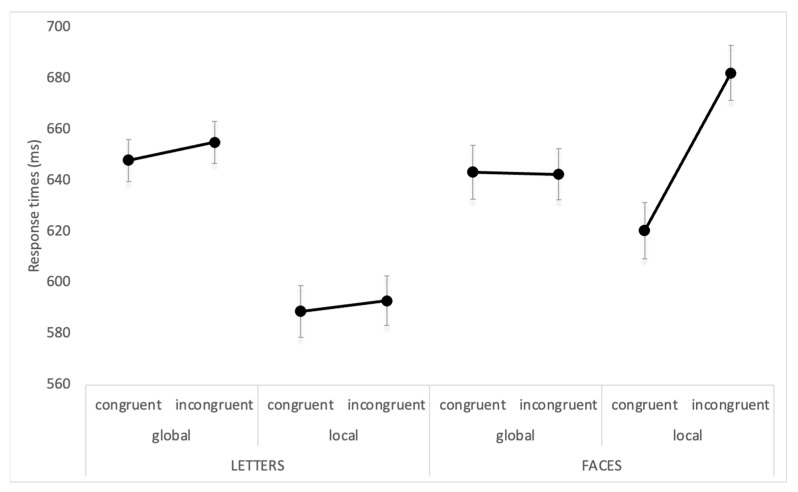
Average reaction times expressed in ms for letters and faces in the different experimental conditions and tasks (Experiments 1a and 1b). Bars represent standard errors.

**Figure 4 brainsci-13-00491-f004:**
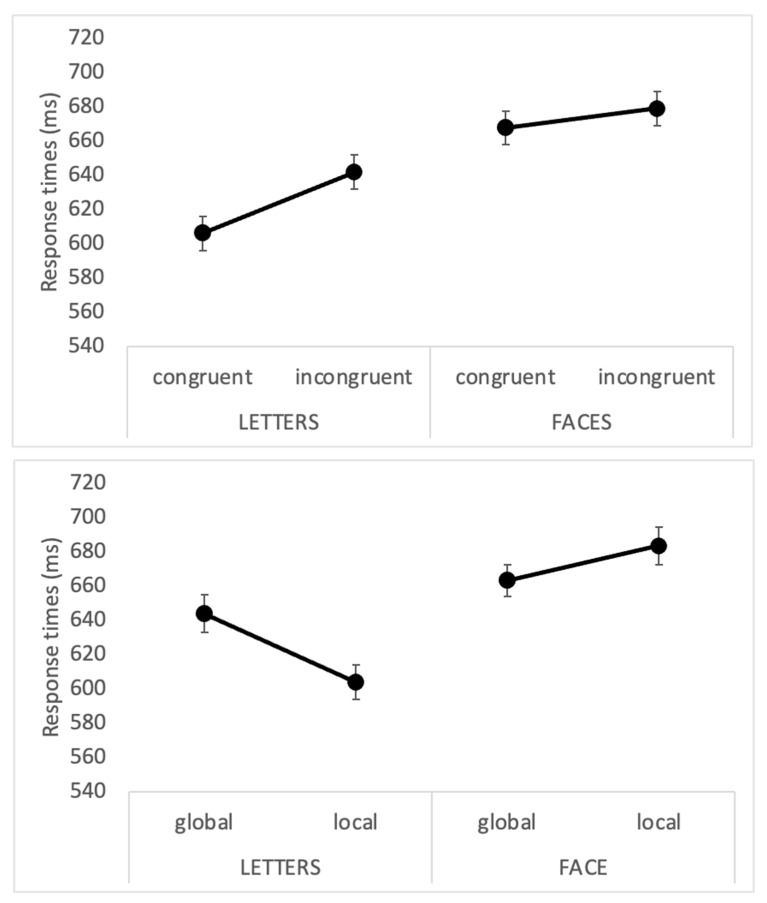
Experiment 2. The results of the two interactions are reported: stimulus x congruency in the upper panel and stimulus x task in the lower panel.

**Figure 5 brainsci-13-00491-f005:**
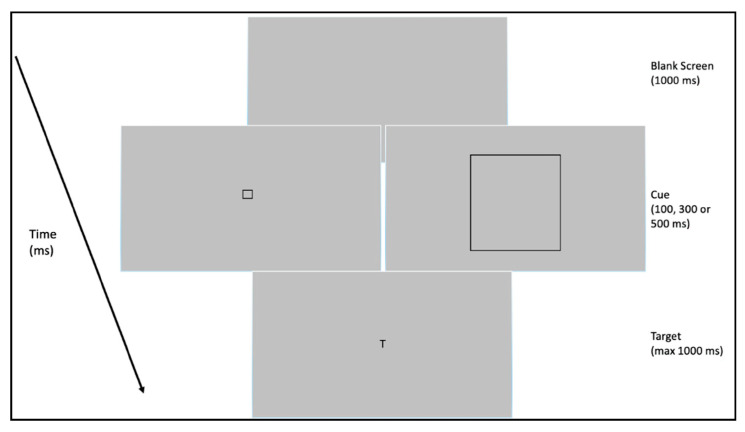
Example of the trial procedure in Experiment 3. After a blank screen (1000 ms), the target appearance could be preceded by no cue, a red dot, a small cue, or a big cue (in the interests of readability, in the figure, only the last two conditions are shown). After 100, 300, or 500 ms, the target (T) was presented, and the participant was asked to respond as quickly as possible by pressing the space bar on a keyboard.

**Table 1 brainsci-13-00491-t001:** Main results of Experiment 2 in terms of response times (ms) for the processing of faces and letters in the global and local conditions for congruent and incongruent stimuli.

		FACES		LETTERS	
		Global	Local	Global	Local
RTs	Congruent	655	681	620	592
	Incongruent	672	686	668	616

**Table 2 brainsci-13-00491-t002:** Mean reaction times for Experiment 3 in response to targets preceded by no cue or a dot, a big square, or a small square as cues. Results are reported for SOAs of 100, 300, and 500 ms. Values in brackets indicate standard deviations.

	No Cue	Dot	Big Square	Small Square
SOA = 100	380 (73)	364 (82)	355 (84)	348 (78)
SOA = 300	369 (69)	313 (79)	331 (80)	317 (86)
SOA = 500	371 (71)	316 (80)	331 (85)	328 (77)

**Table 3 brainsci-13-00491-t003:** Correlational data between the cue-size effect at different SOAs (100, 300, and 500 ms) and the local advantage for letters and faces.

	Cue-Size Effect	Cue-Size Effect	Cue-Size Effect
	SOA = 100	SOA = 300	SOA = 500
Local advantage for letters (=global RT − local RT)	*R* = −0.1, *p* = 0.5	*R* = 0.13, *p* = 0.4	*R* = 0.3, *p* = 0.04 *
Local advantage for faces (=global RT − local RT)	*R* = −0.2; *p* = 0.18	*R* = −0.02; *p* = 0.9	*R* = 0.16, *p* = 0.3

Note. * refers to statistical significance.

## Data Availability

Data can be asked to the corresponding author.

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
