# Peer review of "Global and Local Processing of Letters and Faces: The Role of Visual Focal Attention"

_brainsci, 2023, doi:10.3390/brainsci13030491_

Round 1
Reviewer 1 Report
I find it difficult to compare data across experiments either because of inconsistencies in the way the data are presented, or because the actual data are missing and what is presented is simply the results of statistical tests on the data.
In my opinion, the following information should also be included:
1. The results for Experiment 2 should be also be shown in a format that is identical to Figure 3 (Figure 4 can still remain). When I mentally try to replot the data from expt 1 in a format similar to that used for expt 2, they appear to be very similar qualitatively. But the conclusions from the two experiments are different. The additional figure would help the reader to visualise the differences in the data from the two experiments.
2. Experiment 3 does not report the summary RT data, nor are there any figures to help the reader visualise the data. Figures plotting RT and the speed indices mentioned would be useful in helping the reader gain a quantitative understanding of the correlations described. The R and p values, on their own, without the supporting data, are not at all useful in convincing the reader that the conclusions are correct.
3. The results for expt 1 show some comparison of mean RTs (e.g. line 205: "593 vs 682ms; p<0.0001"). I presume these are t-tests. If they are, that could be explicitly stated.
If these are fixed, it would make things easier for the reader.
Author Response
I find it difficult to compare data across experiments either because of inconsistencies in the way the data are presented, or because the actual data are missing and what is presented is simply the results of statistical tests on the data.
In my opinion, the following information should also be included:
- The results for Experiment 2 should be also be shown in a format that is identical to Figure 3 (Figure 4 can still remain). When I mentally try to replot the data from expt 1 in a format similar to that used for expt 2, they appear to be very similar qualitatively. But the conclusions from the two experiments are different. The additional figure would help the reader to visualise the differences in the data from the two experiments.
Response: We agree with the reviewer that, in principle, the plots of the two experiments should be the same. We report below the plot of Experiment 2 in the same format used for the data of Experiment 1. We believe that this plot is, instead, misleading for the reader and does not allow a clear understanding of the data, because it does not allow to focus on the main statistically significant results. Conversely, the figure 4 used in the manuscript, plots the significant interactions helping the reader in focusing on the main results. However, if the reviewer believes that this figure might be necessary for a better reading of the results, we will include it in the manuscript.
- Experiment 3 does not report the summary RT data, nor are there any figures to help the reader visualise the data. Figures plotting RT and the speed indices mentioned would be useful in helping the reader gain a quantitative understanding of the correlations described. The R and p values, on their own, without the supporting data, are not at all useful in convincing the reader that the conclusions are correct.
Response: We followed the reviewer comments and we report the results of Experiment 3 in (new) Table 1 (lines 357-364).
- The results for expt 1 show some comparison of mean RTs (e.g. line 205: "593 vs 682ms; p<0.0001"). I presume these are t-tests. If they are, that could be explicitly stated.
Response: Yes, the reviewer is correct and we have now included the detail in the manuscript (lines 205-207).
If these are fixed, it would make things easier for the reader.

Reviewer 2 Report
Overall, the manuscript "Global and Local Processing of Letters and Faces: The Role of Visual Focal Attention" presents a well-designed study investigating the global and local processing of letters and faces, and the attentional mechanisms that might be linked to eventual differences. The study uses hierarchical, congruent and incongruent letters and faces in different time conditions (180 and 500 ms) and the results are clearly and concisely presented.
The study's findings support the hypothesis that the preference towards local or global processing can vary according to different characteristics of the stimuli involved, such as stimulus type and stimulus time duration. The results show that letters are processed more efficiently at the local level when presented for 180 ms, with a reversal of the global interference effect. Conversely, faces are still processed more efficiently at the global level, with evidence of global advantage and global interference. When the same stimuli are presented for longer (500 ms), letters are still processed more efficiently at the local level, but there is still a tendency, even if not significant, toward a global processing advantage for faces.
The study also measures the cue-size effect, which is the ability to modulate visual focal attention based on the characteristics of the cue. The results show that the cue-size effect has a statistically significant correlation with the local processing advantage for letters but not for faces. This suggests that the stimulus nature and the cognitive processes it elicits are crucial in the preferential global vs local processing and that focal attention can be modulated by exogenous factors, such as cue size, but in different ways depending on the stimulus type.
Some minor corrections that could be made include:
-Providing more detailed information about the stimuli used in the experiments, such as the specific font used for the letters and the source of the face images.
-Providing more detailed information about the experimental procedures, such as the exact instructions given to the participants and the specific buttons they were asked to press to respond.
-Clarifying the statistics used to analyze the data, such as the specific test used to determine significance and the exact p-values reported.
One potential area for improvement could be to include a control group that uses a neutral stimulus, such as a geometric shape, to further support the conclusion that the observed effects are specific to letters and faces. Additionally, it would be beneficial to include more detailed information about the cue-size effect, such as the specific size of the cue and how this relates to the size of the target stimulus.
The papers “Parietal Representations of Stimulus Features Are Amplified during Memory Retrieval and Flexibly Aligned with Top-Down Goals,” "Reliability of oculometrics during a mentally demanding task in young and old adults," “Task context impacts visual object processing differentially across the cortex,” and "Using eye movement analysis to study auditory effects on visual memory recall" could be relevant references for this paper as they all deal with attentional processes, which are related to the topic of the paper.
The discussion does mention some limitations of the study, such as the use of only two stimulus durations and the use of only a few specific stimuli. However, it does not specifically address the sample size or the participant groups used in the study. It would be beneficial for the authors to address the potential impact of the sample size on the generalizability of the results and to provide more information about the characteristics of the participant groups.
The discussion also does not address the difference in the sample size and demographic characteristics of the two experiments. It's important to mention that the sample size and the demographic characteristics of the two experiments should be matched as much as possible to avoid any bias in the results.
In addition, the authors could include more discussion on the implications of the findings for real-world applications, such as how the observed differences in global and local processing of letters and faces might impact reading or face recognition tasks, and how the sample size and demographic characteristics of the participants might influence these results.
Overall, the manuscript is well-conducted and well-written, but a little more detail in certain areas and a bit more discussion on the implications of the findings would be beneficial.
Author Response
Overall, the manuscript "Global and Local Processing of Letters and Faces: The Role of Visual Focal Attention" presents a well-designed study investigating the global and local processing of letters and faces, and the attentional mechanisms that might be linked to eventual differences. The study uses hierarchical, congruent and incongruent letters and faces in different time conditions (180 and 500 ms) and the results are clearly and concisely presented.
The study's findings support the hypothesis that the preference towards local or global processing can vary according to different characteristics of the stimuli involved, such as stimulus type and stimulus time duration. The results show that letters are processed more efficiently at the local level when presented for 180 ms, with a reversal of the global interference effect. Conversely, faces are still processed more efficiently at the global level, with evidence of global advantage and global interference. When the same stimuli are presented for longer (500 ms), letters are still processed more efficiently at the local level, but there is still a tendency, even if not significant, toward a global processing advantage for faces.
The study also measures the cue-size effect, which is the ability to modulate visual focal attention based on the characteristics of the cue. The results show that the cue-size effect has a statistically significant correlation with the local processing advantage for letters but not for faces. This suggests that the stimulus nature and the cognitive processes it elicits are crucial in the preferential global vs local processing and that focal attention can be modulated by exogenous factors, such as cue size, but in different ways depending on the stimulus type.
Some minor corrections that could be made include:
-Providing more detailed information about the stimuli used in the experiments, such as the specific font used for the letters and the source of the face images.
Response: We have now included the requested details in the manuscript, method section (lines 152, 182-183).
-Providing more detailed information about the experimental procedures, such as the exact instructions given to the participants and the specific buttons they were asked to press to respond.
Response: We have now included the requested details in the manuscript, method section (lines 169-172, 194-198).
-Clarifying the statistics used to analyze the data, such as the specific test used to determine significance and the exact p-values reported.
Response: The requested information has been added in the manuscript (lines 202-207).
One potential area for improvement could be to include a control group that uses a neutral stimulus, such as a geometric shape, to further support the conclusion that the observed effects are specific to letters and faces. Additionally, it would be beneficial to include more detailed information about the cue-size effect, such as the specific size of the cue and how this relates to the size of the target stimulus.
Response: We thank the reviewer for the suggestion and we completely agree with it. Unfortunately, this further condition cannot be implemented in the present study since it is not possible to administer the control condition to the same participants who have taken part in the other two experiments. Nonetheless we are currently working on this and hope to get and publish the results as soon as possible.
We have now better clarified that the cue sizes matched the stimulus sizes. In particular, the small cue had the same size of the local element in the hierarchical stimuli (both faces and letters) and the large cue had the same size of the global element in the hierarchical stimuli (both faces and letters) – lines 288-290.
The papers “Parietal Representations of Stimulus Features Are Amplified during Memory Retrieval and Flexibly Aligned with Top-Down Goals,” "Reliability of oculometrics during a mentally demanding task in young and old adults," “Task context impacts visual object processing differentially across the cortex,” and "Using eye movement analysis to study auditory effects on visual memory recall" could be relevant references for this paper as they all deal with attentional processes, which are related to the topic of the paper.
Response: As suggested by the reviewer, we included the following paper in the discussion section (lines 442-444) and now it is listed as reference #40.
The discussion does mention some limitations of the study, such as the use of only two stimulus durations and the use of only a few specific stimuli. However, it does not specifically address the sample size or the participant groups used in the study. It would be beneficial for the authors to address the potential impact of the sample size on the generalizability of the results and to provide more information about the characteristics of the participant groups.
Response: We have followed the reviewer suggestion and we included the limitation of the study in the discussion (lines 445-450).
The discussion also does not address the difference in the sample size and demographic characteristics of the two experiments. It's important to mention that the sample size and the demographic characteristics of the two experiments should be matched as much as possible to avoid any bias in the results.
Response: Demographic features of the two groups of participants were matched between experiments 1 and 2 (t-test comparisons; all p>0.1) – lines 236-238.
In addition, the authors could include more discussion on the implications of the findings for real-world applications, such as how the observed differences in global and local processing of letters and faces might impact reading or face recognition tasks, and how the sample size and demographic characteristics of the participants might influence these results.
Response: We included a paragraph in the discussion highlighting the impact of the demographic features of the specific participant sample involved in the study. We also described the neuropsychological applications of the present results, suggesting the potentiality in investigating the cognitive mechanisms for a better understanding of simultanagnosia and prosopagnosia (lines 445-450).
Overall, the manuscript is well-conducted and well-written, but a little more detail in certain areas and a bit more discussion on the implications of the findings would be beneficial.
Response: thank you very much for your feedback. We hope we have addressed the reviewer’s concerns with the present review.

Reviewer 3 Report
This paper attempts to explore the attention mechanism, from local to global or from global to local, on letters and artificial face images. This is an interesting and important topic because the recent dominant artificial intelligence techniques, deep learning (listed below), are inspired by human attention processings. The finding of this work may inspire future work in AI areas. Overall, this is a good paper, although there are several issues to addess.
Issues:
1. The Introduction MUST be well organized. It's not friendly to readers putting thousands of words in ONE paragraph.
2. As above-mentioned, this paper actually closely related to the current AI techniques. Thus, great advance in various AI areas should be presented to better show the importance of this paper. Below I list several typical works, where deep learning model building bricks are designed based on attention mechnism:
. Attention is all you need
. On the Relationship between Self-Attention and Convolutional Layers
. Robust visual tracking via scale-and-state-awareness
. Multi-Attention Network for Compressed Video Referring Object Segmentation
. Attention Augmented Convolutional Networks
. Siamese local and global networks for robust face tracking
. Neighbor-view enhanced model for vision and language navigation
. HOP: History-and-Order Aware Pre-training for Vision-and-Language Navigation
. Deep Residual Learning for Image Recognition
. Hedged deep tracking
. Hierarchical Modular Network for Video Captioning
. Overwater image dehazing via cycle-consistent generative adversarial network
. Image editing with varying intensities of processing
. V2C: Visual Voice Cloning
. Release the power of online-training for robust visual tracking
3. All figures should be resize to a proper size making the fonts in figures have similar size as the font used in body text.
4. A clearer statement in the conclusion should be given. Tell readers what you exactly found.
Author Response
This paper attempts to explore the attention mechanism, from local to global or from global to local, on letters and artificial face images. This is an interesting and important topic because the recent dominant artificial intelligence techniques, deep learning (listed below), are inspired by human attention processings. The finding of this work may inspire future work in AI areas. Overall, this is a good paper, although there are several issues to address.
Issues:
- The Introduction MUST be well organized. It's not friendly to readers putting thousands of words in ONE paragraph.
Response: As suggested by the reviewer the introduction has been better organized by splitting large paragraphs in smaller ones.
- As above-mentioned, this paper actually closely related to the current AI techniques. Thus, great advance in various AI areas should be presented to better show the importance of this paper. Below I list several typical works, where deep learning model building bricks are designed based on attention mechnism:
. Attention is all you need
. On the Relationship between Self-Attention and Convolutional Layers
. Robust visual tracking via scale-and-state-awareness
. Multi-Attention Network for Compressed Video Referring Object Segmentation
. Attention Augmented Convolutional Networks
. Siamese local and global networks for robust face tracking
. Neighbor-view enhanced model for vision and language navigation
. HOP: History-and-Order Aware Pre-training for Vision-and-Language Navigation
. Deep Residual Learning for Image Recognition
. Hedged deep tracking
. Hierarchical Modular Network for Video Captioning
. Overwater image dehazing via cycle-consistent generative adversarial network
. Image editing with varying intensities of processing
. V2C: Visual Voice Cloning
. Release the power of online-training for robust visual tracking
Response: We thank the reviewer for the appreciation of our study in terms of potential influence it can have on AI literature. Nonetheless, this is not within our field of expertise and we do not feel confident to explicitly discuss such implications within the paper.
- All figures should be resize to a proper size making the fonts in figures have similar size as the font used in body text.
Response: Thank you for the suggestion. We will make sure that in the final and (hopefully) published version of the paper, figures will be the proper size.
- A clearer statement in the conclusion should be given. Tell readers what you exactly found.
Response: Thank you for the suggestion. We have now included a final statement in the conclusion (lines 466-470).

Round 2
Reviewer 3 Report
The writing and organization of the revised version are better than the last one. However, there exist some crucial issues in the experiments which may lead to misleading conclusions. First, there are only 23 participants and Male:Female=6:17. Such a small-scale and unbalanced data has a high risk of bias in the collected data. Additionally, only 3 out of 26 letters are used for experiments, which is insufficient to give a fair conclusion. As these are the basis of the whole paper, I think a major revision is not enough to fix these issues and suggest preparing a new manuscript.
Author Response
Dear editor
Thank you for giving us the opportunity to revise our manuscript. As you suggested, the following changes have been implemented to the manuscript:
Results of Experiment 2 indicate two statistically significant interactions (stimulus x congruency and stimulus x task) which, we believe, are cleanly shown in the two panels of Figure 4. Such results would not be shown by a figure compacting result, as the one used for the results of Experiment 1. However, we understand the necessity to have a full picture of the data in order to easily compare the results between Experiments 1 and 2. Accordingly, we have added a Table in the manuscript (now Table 1) where we report all the results of Experiment 2 that were reported in the Figure included in the previous response letter (round 1 of revisions). This ensures both the completeness of data presentation and the possibility to make direct comparisons between the two experiments (lines 261-265).
As suggested by the Editor, a paragraph addressing the limitations of the study has been added in the discussion. In particular, we acknowledge that the participant sample is limited in terms of demographic features and number. Furthermore, we also recognize that future study will be aimed at generalizing the results of the study, by including one or more control condition adopting different typologies of stimuli, such as geometric shapes. This will strengthen and further support the conclusion that the observed effects are specific to letters and faces. (lines 442-460).
We hope that the manuscript can now be accepted for publication in Brain Sciences.
Kind regards
Lisa Arduino
